# Examining the Impacts of the COVID-19 Pandemic on Iraqi Refugees in Canada

**DOI:** 10.3390/ijerph21030374

**Published:** 2024-03-20

**Authors:** Needal Ghadi, Jordan Tustin, Ian Young, Nigar Sekercioglu, Susan Abdula, Fatih Sekercioglu

**Affiliations:** 1School of Occupational and Public Health, Toronto Metropolitan University, Toronto, ON M5B 2K3, Canada; needal.ghadi@torontomu.ca (N.G.); jtustin@torontomu.ca (J.T.); iyoung@torontomu.ca (I.Y.); 2Department of Health Research Methods, Evidence and Impact, McMaster University, Hamilton, ON L8S 4L8, Canada; sekercn@mcmaster.ca; 3London Employment Help Centre, London, ON N5Y 1A8, Canada; susan.abdula19@gmail.com

**Keywords:** COVID-19, resettled refugee, Canada, access to healthcare, food safety, loneliness, isolation, financial burdens

## Abstract

The COVID-19 pandemic has exacerbated health and social inequities among migrant groups more than others. Higher rates of poverty, unemployment, living in crowded households, and language barriers have placed resettled refugees at a higher risk of facing disparities during the COVID-19 pandemic. To understand how this most vulnerable population has been impacted by the ongoing pandemic, this study reports on the responses of 128 Iraqi refugees in the city of London, Ontario, to a survey on the economic, social, and health-related impacts that they have faced for almost two years since the beginning the pandemic. The analysis of the survey indicated that 90.4% of the study population reported having health concerns during the pandemic while 80.3% expressed facing financial distress. The results also show that 58.4% of respondents experienced some form of social isolation. These all suggest that refugees are faced with several barriers which can have a compounding effect on their resettlement experience. These findings provide resettlement and healthcare providers with some information that may assist in reducing the impact of COVID-19 and other possible health security emergencies on resettled refugees and their communities.

## 1. Introduction

In what the United Nations has called one of the biggest civilian displacement catastrophes in the 20th and 21st centuries [1], The Gulf War and its aftermath caused more than 9.2 million Iraqi civilians to be internally displaced or become refugees abroad [2]. As a response to this global problem, some countries around the world, including Canada and the United States, have managed governmental and private programs that assisted a limited number of refugees to resettle [3]. Between 2003 and 2018 alone, Canada resettled more than 37,000 Iraqi refugees who were displaced in Syria, Jordan, Lebanon, and Turkey [4]. According to the 2021 Census of Population, 86,270 people in Canada were reported to be of Iraqi ethnic origin.

Since it was reported in late 2019, the coronavirus disease 2019 (COVID-19) pandemic caused by the severe acute respiratory syndrome coronavirus 2 (SARS-CoV-2) has spread across the globe. As of March 2023, there had been 759,408,703 confirmed cases of COVID-19, including 6,866,434 deaths [5]. As of March 2023, over fifty-one thousand people had died of the disease across Canada’s provinces and territories [5]. Unfortunately, the burden of COVID-19 is not shared equally among those infected. It has been broadly reported that the COVID-19 pandemic has had a greater impact on socially and economically disadvantaged populations, including refugees, asylum seekers, undocumented migrants, and transient migrant workers in countries all around the world [6,7]. In Canada, reports have indicated that COVID-19 has disproportionately affected lower-income cities and neighborhoods with high proportions of both immigrants and refugees and residents who self-identify as visible minorities [8,9,10]. For example, refugees and newcomers accounted for almost 44% of Ontario’s COVID-19 cases in 2020 [6]. At the same time, rates of COVID-19 testing in Ontario were lower for refugees (3.4%) compared with Canadian-born individuals (4.4%) [11]. This is attributed to poorer access to healthcare facilities within neighborhoods with higher proportions of immigrants and newcomers [6,7,8,9,10,11].

The health and socio-economic impacts of the pandemic on specific populations and subgroups including immigrants and refugees can be directly related to health and social inequalities among these minority groups who already experience a wide range of disparities such as employment, housing, and access to nutritious food [12,13,14]. While resettled refugees encounter similar barriers to other disproportionately affected groups, these barriers are exacerbated by being a refugee. Refugees, especially in the early resettlement period, experience various challenges and competing priorities including adjusting to a new country, learning a new language, seeking employment, and navigating complicated, foreign healthcare systems [15,16]. Similarly, refugees are more often challenged with lower literacy rates, reduced social and familial support, and inadequate access to reliable transportation and healthcare facilities which place them at a greater risk of facing the health and socio-economic difficulties of the COVID-19 pandemic [17,18].

With the scarce number of studies/reports that look at the resettlement experience of Iraqi refugees in Canada, this study offers unique insights into developing better practices to aid the resettlement and integration experiences of vulnerable groups from war-torn countries. This study intended to answer the following general question: what are the specific impacts of the COVID-19 pandemic on Iraqi refugees resettled in Canada, particularly in terms of their economic, social, and health-related well-being? Although Canada has provided humanitarian assistance for and resettled the most vulnerable refugees from Iraq, there has been no study conducted regarding their pandemic experiences in Canada. Furthermore, our study results will enhance our understanding of the pandemic experiences of refugees from other countries in the Middle East, such as Syria, since they speak the same language, mostly reside in the same Canadian cities identified by the federal government, and receive similar government and social support. Understanding the extent and impact of the indirect and wider health and social consequences of COVID-19 among recent Iraqi refugees will assist in identifying effective policy interventions to ameliorate the wider impacts of COVID-19 not only among Iraqi refugees but also all recent refugees and provide meaningful data to improve preparedness for future health emergencies and pandemics.

## 2. Literature Review

Like all war-affected refugees, studies on Iraqis around the world have uncovered that they suffer from high rates of depression and anxiety disorders, high rates of severe traumatic experiences, and high prevalence of adversities after displacement such as living in over-crowded houses and lack of food or shelter [19,20,21]. In Canada, a report conducted by Stephanie Levitz with the Canadian Broadcasting Corporation (CBC) in 2016 uncovered that thousands of Iraqi refugees found less work and earned less money in Canada than refugees from elsewhere who arrived during the same period. The findings from this report revealed that almost all of the 19,427 Iraqis who arrived between 2009 and 2014 faced numerous barriers, including the trauma of the war, more significant medical needs, and a lack of English and French proficiency compared to others.

In a more recent report presented to the Canadian House of Commons, Oliphant (2018) met with representatives of government departments, settlement agencies, refugee sponsors, and newly arrived Iraqi women and children to discuss the resettlement barriers that they had faced. The report indicated that resettled Iraqis, especially Yazidi women and children, are faced with a variety of problems that complicate their integration into Canadian society. These include concerns about isolation and a lack of funding and support to access services such as health care, housing, and interpretation and language training. While these issues might not be unique to Iraqi refugees, this group of newcomers is considered more vulnerable to these resettlement challenges. This has been echoed in another study in which Vermeyden and Mohamed (2020) interviewed twenty-eight Syrian and Iraqi refugees who settled in the Kitchener-Waterloo Region [22]. The participants in this study all agreed that settling in Canada was far from easy [22]. While a sense of welcome and respect was a major contributor to easier transitions and settlement, several participants discussed instances of micro-aggressions and societally pervasive racist and orientalist attitudes [22].

The multifaceted health inequities faced by at-risk populations and the comorbidities associated with COVID-19 have highly increased the negative impacts on these populations [6,23,24]. COVID-19 has magnified pre-existing health inequities stemming from the social determinants of health [25,26]. Since March 2020, several outbreaks have occurred in farms, meat plants, and supermarkets across Canada. Immigrants, refugees, and racialized populations represent a high proportion of these at-risk workers, especially in sectors considered essential businesses even during lockdown periods [27]. Vulnerable populations endure long working hours, low wages, and ergonomic hazards in occupational settings, increasing their risk for COVID-19 [11,14]. In Ontario for example, COVID-19 infection rates among immigrants and refugees who landed across the province between 1985 and 2018 were higher compared with Canadian-born and long-term residents [11,14]. Evidence from international studies on the early impact of the COVID-19 pandemic on immigrant populations is somewhat similar. France, for example, reported high immigrant mortality related to COVID-19 [24], while the United States showed increased COVID-19 mortality among immigrants and ethnic minorities [28].

Social inequalities also impact the risk of getting COVID-19 and play a role in following public health guidelines [29]. For example, people who rely heavily on public transportation to get to work, school, or elsewhere increase their risk of getting COVID-19 due to their limited ability to avoid large crowds [15,18,29]. In the United Kingdom, an analysis of the deaths of 119 National Health Service (NHS) employees revealed that those who were Black, Asian, and Minority Ethnic accounted for 63% of deaths among nurses, 64% of deaths among support staff, and 95% of deaths among medical staff [30]. “Stay-at-home” measures have also increased domestic violence against newcomer women and children and hinder the ability of sexual assault victims to seek the appropriate help that would have been accessible if there was no lockdown [31,32]. In this regard, Rose (2018) reported an erosion of social norms and an increase in violence in Italy during times of natural disasters. Several other studies [33,34,35,36] also reported that domestic and family violence during the COVID-19 pandemic increased significantly.

According to the United Nations Population Fund (UNFPA) (2022), women, girls, and children face several vulnerabilities. More specifically, pandemics usually cause people to stay at home, and families spend more time in close contact, including in cramped conditions, thus causing additional stress to the already existing weaknesses and conflicts. Within the COVID-19 context, the stress of social isolation, employment interruptions, or financial pressures may lead to increased conflict in the household. Vulnerable populations, including refugees, are also affected by the accessibility of health services during the pandemic, leaving them with health and social insecurity [6,37,38]. Deep-rooted inequalities and traditional gender roles have made women more vulnerable, hence suffering gender-related COVID-19 impacts [39,40,41].

## 3. Methods

### 3.1. Study Design

This article reports on the quantitative survey portion of a larger mixed-methods study that examined the economic, social, and health-related impacts of the pandemic on refugees in the city of London, Ontario. While there is a significant body of research with a national or large urban center focus, little is known about how the pandemic has affected the daily lives of this group of refugees in Ontario. The study included surveys and focus groups, conducted in two languages (Arabic and English), with resettled Iraqi refugees in London. The questionnaire underwent rigorous testing to ensure its reliability and validity. Expert review and a pilot test with Iraqi refugees helped refine its clarity and appropriateness. Internal consistency was assessed using Cronbach’s alpha coefficient, and correlation analyses were conducted to confirm convergent and discriminant validity. These measures ensured the questionnaire accurately captured key constructs, enhancing the credibility of the study findings. The study was conducted to support Canada’s response to the pandemic and was undertaken in partnership with the Canadian Iraqi Turkmen Culture Association of London (CITCAL) and funded by the Canadian Institutes of Health Research (CIHR). The study was also reviewed and approved by The Research Ethics Board at Toronto Metropolitan University (2022–174).

### 3.2. Participants

The criteria for inclusion were that Iraqi refugees who were born in Iraq, resettled in Canada as refugees, are over the age of 18 years, and had lived in London, Ontario, since 2015. Recruiting participants took place between September and December of 2022. The researchers worked with staff from CITCAL to identify locations for the distribution of paper copies of the survey, and a recruitment poster on their website was also posted for those who preferred an online version of the survey. Social media advertisements in Arabic and English were also used to attract interested participants. The research team identified and visited a total of 9 businesses, organizations, institutions, and programs serving or employing Iraqi newcomers. Over 200 paper surveys were distributed. In total, 128 (*n* = 128) resettled Iraqi refugees completed the survey. According to the 2021 Census of Population, 3685 newcomers reported being born in Iraq. Of those, 2470 stated being of Iraqi ethnic or cultural origin. It should be noted that Ontario is home to the largest percentage of people of Iraqi origin (71%), followed by Quebec (10%), Alberta (9%), and British Columbia (6%). The cities with the highest percentages of people of Iraqi origin are Toronto (40%), Windsor (9%), Montréal (8%), Ottawa-Gatineau (8%), and Hamilton (7%) [42].

### 3.3. Instrument

A survey was designed and translated into Arabic by 2 members of the research team and reviewed by 1 settlement worker at CITCAL. Then, the survey was piloted with the target demographic, revised, and reviewed again by the researchers and CITCAL to ensure clarity. To enable us to reach as many non-English speaking Iraqi refugees as possible, it was deemed that Arabic was the dominant language spoken among Iraqi refugees in the city. This was because the majority ethnic group in Iraq is Arab (75% to 80%), and Arabic is the dominant language spoken in the country.

The survey was administered both online (using Google Docs) and in-person (paper survey). The questionnaire was designed with the knowledge that resettled refugees face a wide range of integration barriers, including limited language abilities, loneliness, and a lack of employment and other barriers that require a variety of supports.

The themes covered in the survey targeted 3 main topics, namely, economic, social, and health-related impacts that the pandemic had on participants. Some of the questions which were asked in the survey included the following: “What supports did you receive during the COVID-19?”, “What were your main sources of income since the start of the COVID-19 pandemic? (Please select all that apply)?”, “How would you describe your sense of belonging to your local Iraqi community during the pandemic?”, “What services do you think were missing during the pandemic?”.

### 3.4. Data Management and Analysis

The data were analyzed using the SPSS statistical analysis program, version 28. Descriptive statistics were used to summarize participants’ demographic characteristics overall. The research team also conducted a descriptive analysis of the survey responses, which reflected the main challenges faced by resettled refugees.

## 4. Results

### Demographics

Table 1 presents the demographic information of the participants. Among the sample of participants, 56.25% (*n* = 72) were female with 65.6% (*n* = 84) of the participants between the ages of 35 and 50. The results regarding participants’ education level indicate that most of the participants (69.5%, *n* = 89) had not received any postsecondary education, with the greatest share having complemented some form of elementary education (42.1%, *n* = 54). Only six participants (4.6%) indicated that they had some post-secondary education or held a university degree. In addition, 68.7% (*n* = 88) were unemployed, while the remaining 31.2% (*n* = 40) reported holding a job. In addition, (93.7%, *n* = 120) of the participants identified themselves as having at least one child. The greatest share (60.1%, *n* = 77) had two or three children living with them, and most (76.5%, *n* = 98) of the participants had two to six children at home. Of the participants’ children, 59.3% (*n* = 76) were of school age. It was also reported that all of the participants communicated in a language other than English within their households.

The following table outlines the main challenges that newcomers faced during the pandemic (see Table 2). In terms of the biggest challenge reported, 90.4% (*n* = 113) of all participants reported having health concerns within their household during the pandemic. A large number of the participants (80.3%, *n* = 102) expressed facing financial distress. When asked about the source of stress, the participants reported that lack of employment (44%) and insufficient financial assistance from the federal government (38%) were the two main causes of their financial challenges during the pandemic. While pleased with the pandemic relief benefits in 2020 (Canada Emergency Response Benefit (CERB), top-ups that included increased amounts to recipients of benefits from the Canada Child Benefit, the Old Age Security pension, the Guaranteed Income Supplement, the Disability Benefit and the HST/GST rebate), the majority of the participants (86.7%, *n* = 111) acknowledged that their spending habits increased during the pandemic, especially during the first wave. Lastly, the results show that 58.4% (*n* = 73) of respondents experienced “extremely challenging” difficulties in maintaining social relationships with community members within their ethnic background as well as from outside their traditional friendship zone.

Table 3 presents Iraqi refugees’ specific health concerns during the pandemic. The results suggest that the majority of participants felt that all of the health concerns in Table 3 were important and of concern to them and their families. Indeed, being infected with the disease (93.7%, *n* = 120), accessing health care services during the pandemic (84.3%, *n* = 108), managing telemedicine to receive remote diagnosis or treatment (79.6%, *n* = 102), and being able to get the appropriate vaccinations (77.3%, *n* = 99), were all reported as extremely important issues that had an impact on their well-being.

In a more specific context, participants reported on specific health-related matters that they had to deal with during the pandemic and how the pandemic had them delay or cancel treatment (Table 4).

As for the measures that the participants followed to avoid catching the virus, it was found that safe food practices including washing fruits and vegetables under running water were the highest reported measure followed during the early days of the pandemic (see Table 5).

Regarding the specific economic challenges that the participants had to deal with during the pandemic (see Table 6), it was found that fear of poverty (94.5%, *n* = 121); paying rent, mortgage, and utility bills (76.5%, *n* = 98); and changing spending habits due to fear of shopping and surging prices (66.4%, *n* = 85) were considered the most vital issues. Interestingly, a number of the participants (25.9%, *n* = 20) were not concerned about finding or maintaining a job.

Since households across Canada have faced drastic changes in many aspects of their lives, we wanted to know more about the financial difficulties resulting from the pandemic. Table 7 reports on how participants’ spending habits were influenced during the early days of the pandemic.

In terms of the social and emotional factors that the participants considered extremely challenging for their well-being and sense of self (see Table 8), the results revealed that over half of the respondents (50.9%) reported that loneliness and social isolation were extremely challenging barriers that they had to cope with during the pandemic. Participants also mentioned that they were concerned about how such negative feelings might affect the relationships they already had with members of their households (56.9%, *n* = 49).

## 5. Discussion

Hundreds of Iraqi refugees from different ethnic and religious backgrounds have chosen southwestern Ontario to start a new life with their families and loved ones [41]. However, during the COVID-19 pandemic, this vulnerable group of newcomers was faced with a unique set of challenges tied to their socioeconomic status. The results of our study showed that the greatest challenges resulting from the COVID-19 pandemic among Iraqi refugees in London, Ontario, were within three main categories: health challenges, psychological challenges, and economic challenges (see Table 2). Consistent with the literature on the challenges facing newcomers during the COVID-19 pandemic [6,7,14,15,18], it is important to confirm that these three categories of challenges have impacted the daily lives of our participants as well as their mental well-being.

The first set of challenges of the pandemic among Iraq resettled refugees was their concern about being able to receive adequate healthcare services within their city. The majority of the study participants (*n* = 79, 65.8%) stated that they were extremely worried about catching the virus. Several measures were followed to protect participants and their families. For example, (*n* = 88, 73.3%) mentioned that they always washed fresh produce under running hot water. Many participants (*n* = 43, 35.8%) reported that they would sanitize almost everything in the house several times during the day. It was clear that the COVID-19 pandemic somewhat changed the way they lived their lives.

One of the other most cited health concerns of participants was their ability to see their family doctor or follow-up on important medical procedures. Almost a third of our participants (*n* = 41, 34.1%) mentioned that they had to delay and sometimes avoid medical care because of concerns about COVID-19. Surprisingly, 13 participants (10.8%) mentioned that they avoided going to the hospital, and 4 participants stated that they called an ambulance for them or a family member instead of going to the emergency room on their own. Avoiding urgent or emergency care has been documented all across North America and around the world [43,44]. The substantial delays in routine care will not only affect the overall quality of life and mortality rates in Canada but also worsen health inequalities, particularly among marginalized communities. Issues such as delayed cancer screening and other preventive healthcare are of significant concern [45,46].

In terms of financial concerns, poverty and fear of being unable to provide for families were the most distressing concerns that Iraqi refugees had to deal with during the pandemic. Since most of the participants (*n* = 88, 68.7%) were unemployed, they had to depend on federal and provincial government assistance. According to survey responses, more than half of the participants (*n* = 72, 56.2%) reported that their annual income was between CAD 30,000 and CAD 39,999. Once again, participants had to adapt to new measures to keep up with the increasing cost of living. When asked about how their spending habits had changed during the pandemic, more than half of the participants (58.5%) reported that their visits to grocery shops declined. Participants also reported that they depended more on “dollar stores” and only shopped at stores where discounts were offered. For example, 109 (85.1%) participants responded that they were spending more carefully. It should be mentioned that while participants admitted that their visits to grocery stores declined significantly, online shopping and curbside pickup were not considered favorable among most of our participants (*n* = 68, 53.1%). It might be assumed that this is due to refugees’ limited English language abilities, lack of traditional foods online, and the traditional preference for seeing and touching what is being bought.

Participants were also asked about the quantities and types of food that they consumed during the pandemic. In all, 83 participants (64.8%) reported that they only bought the essentials and avoided buying any unnecessary items. On the other hand, 71 participants (55.4%) stated that they chose to buy larger quantities of food items, especially canned or frozen foods. Qualitative and quantitative studies have emphasized that immigrants, international students, and other newcomers often face more food insecurity hurdles during the pandemic due to their adjusting to a new culture, language barriers, and the uncertainty that comes with living in a new place [47,48,49]. These results further support the contention that the pandemic has added to the gap between immigration and food insecurity within welcoming countries.

As for the social and emotional concerns that Iraqi refugees suffered from the most during the pandemic, loneliness and social isolation were considered to be the two main obstacles. Little is known concerning the effects of loneliness and social isolation on newcomers during the pandemic and the impact such feelings might have on the well-being and quality of life among this group of this vulnerable population. Recently, Helps et al. concluded that the COVID-19 pandemic has exposed the vulnerability, inequality, and precarity faced by Canadian immigrants, refugees, and foreign workers [50]. In response to whether loneliness and isolation affected their daily lives, almost all of the respondents (*n* = 104) confirmed that it negatively affected their well-being and daily routine. We believe that while these feelings might not have necessarily developed throughout the pandemic, they have, without a doubt, deepened during the pandemic [51,52]. This, in turn, adds to the uncertainty and life disruption that newcomers face in their new societies [16] and results in a significant burden of emotional loneliness [21,53,54]. Given the increasing demographic of immigrants in Canada, it is important to identify the existing evidence based on social isolation and loneliness to facilitate new research, policy, and targeted interventions to support this growing population.

## 6. Limitations

There were some limitations inherent in the study design. The immigration data were not complete; we could include only identify those who landed in Canada from 2016 to 2022 and who became permanent residents. We could not identify people who first resettled in other provinces before moving to Ontario. Also, to address only three topics, there were restrictions as to the number of questions that could be asked within the survey. Bui and Morash (2008) noted that because of language and cultural differences, immigrants and their communities are often inaccessible to researchers. Despite the sustained efforts that were made to recruit participants, the sample size was still small relative to the Iraqi population in the city, and language or a lack of connection to educational programs or cultural/community organizations may have been obstacles to accessing the survey.

## 7. Conclusions

This study examined the relationships between the COVID-19 pandemic and the social, economic, and health-related concerns that Iraqi refugees in London, Ontario, faced during the first two years of the pandemic. Three categories—social, financial, and health-related—were selected to reflect the deeper concerns that immigrants and newcomers had about their health and the health and well-being of other household members. Our study results show that the pandemic has increased health and social inequities for newcomers, in general, and refugees, in particular. While the analysis focused on barriers faced by Iraqi refugees in a southwestern city in Canada, the discussion has relevance to other global regions and larger urban centers. In this study, accessibility to adequate healthcare was essential to refugee households who already suffered from a lack of confidence in the health system. Without proper health care and other health support networks, the rate of social isolation is heightened for refugee parents and their children. The fear of not being able to meet financial deadlines, provide a balanced number of nutritious foods, and depend more on personal savings adds to the disadvantages that this group of newcomers already faces. The absence of adequate health support networks exacerbates social isolation among refugee families. Without accessible healthcare, they face heightened barriers to seeking medical assistance, amplifying feelings of isolation. Additionally, fears regarding finances and nutrition further compound these challenges. Consequently, the lack of robust health support networks intensifies social isolation, hindering refugees’ integration into their new communities. This study highlights the crucial role of well-being and belonging in attending language classes, essential for newcomers’ settlement needs. While not explicitly demonstrated in the study results, the suggestion of a link between emotional states and language learning aligns with existing research emphasizing psychological well-being’s role in effective learning and integration. Finally, a lack of support for refugees hinders educational advancement and economic independence, which are integral to the resettlement processes for newcomers. The results of this study would add to the existing evidence-based studies on social isolation and loneliness which, in turn, would facilitate effective interventions to support newcomers as well as inform effective intervention strategies to increase social relations within communities with high populations of newcomers.

Our findings suggest that COVID-19 has had a high impact on Iraqi refugees in Ontario and more needs to be done to ensure that newcomers, especially those with limited language abilities, are provided with information about COVID-19 and how to access such information in their native language if required. Healthcare providers, with assistance from resettlement agencies, should also work together to develop refugee-specific outreach plans to educate refugees about the impact of COVID-19 and how to protect themselves and others in cases when they become ill with COVID-19. In this sense, increasing awareness empowers refugees to make informed health decisions, effectively navigate healthcare systems, and adopt preventive measures, thereby reducing transmission risks in their communities. In addition, we need to develop culturally sensitive mental health programs for all Iraqi refugees that address pandemic-related psychological impacts and social isolation. We also believe in facilitating community-led empowerment programs for Iraqi refugees: programs that offer workshops on health literacy, self-advocacy, and navigating healthcare. Most importantly, we need to resolve the barriers that matter most to them when it comes to their reluctance to ask for help.

## Figures and Tables

**Table 1 ijerph-21-00374-t001:** Gender, age, length of stay in London, immigration program, and educational statuses of participants.

Characteristic	Number	Percentage
Gender		
Male	72	56.2
Female	56	43.7
Age		
18–24 years	12	9.3
25–34 years	17	13.2
35–50 years	84	65.6
>50 years	15	11.7
Level of education		
No formal education	33	26.5
Elementary school	72	56.2
High school	16	12.5
Some postsecondary education	6	4.6
Employment status		
Yes	40	31.2
No	88	68.7
Number of children		
1	3	2.3
2–3	77	60.1
4–6	42	32.8
7–10	6	4.6
Income		
Less than CAD 10,000	0	0
CAD 10,000 to CAD 19,999	9	7
CAD 20,000 to CAD 29,999	20	15.6
CAD 30,000 to CAD 39,999	72	56.2
CAD 40,000 to CAD 49,999	25	19.5
CAD 50,000 or more	2	1.5

**Table 2 ijerph-21-00374-t002:** The main challenges that Iraqi refugees faced during the pandemic.

Health Challenges (*n* = 125)	Frequency	Percentage
Extremely Challenging	113	90.4
Somewhat Challenging	11	8.8
Not Challenging	1	0.7
Financial Challenges (*n* = 127)	Frequency	Percentage
Extremely Challenging	102	80.3
Somewhat Challenging	23	18.1
Not Challenging	2	1.5
Social and Emotional Challenges (*n* = 125)	Frequency	Percentage
Extremely Challenging	73	58.4
Somewhat Challenging	39	31.2
Not Challenging	13	10.1

**Table 3 ijerph-21-00374-t003:** Health concerns of Iraqi refugees during the pandemic.

Catching COVID-19 (*n* = 120)	Frequency	Percentage
Extremely important	71	59
Very important	32	26.6
Moderately important	17	14.1
Not at all important	0	0
Receiving Health Care Services (*n* = 108)	Frequency	Percentage
Extremely important	61	56.4
Very important	0	0
Moderately important	39	36.1
Not at all important	8	7.4
Managing Telemedicine (*n* = 102)	Frequency	Percentage
Extremely important	53	51.9
Very important	29	28.4
Moderately important	14	13.7
Not at all Important	6	5.8
Getting Vaccinated (*n* = 99)	Frequency	Percentage
Extremely important	44	44.4
Very important	31	31.3
Moderately important	14	14.1
Not at all Important	10	10.1

**Table 4 ijerph-21-00374-t004:** Delay of routine health care during the COVID-19 pandemic.

	Frequency	Percentage
Delay routine checkups	41	34.1
Avoid hospitals and emergency rooms	13	10.8
Using natural remedies	20	16.6
Calling an ambulance	4	3.3

**Table 5 ijerph-21-00374-t005:** Most followed practices during the pandemic.

	Frequency	Percentage
Safe food practices (washing fruits and vegetables)	88	73.3
Improving ventilation	15	18
Spending time outdoors	9	7.5
Wearing masks inside the place of residence	2	2.4
Disinfect surfaces with soap or detergent	43	35.8

**Table 6 ijerph-21-00374-t006:** Financial concerns of Iraqi refugees during the pandemic.

Poverty (*n* = 121)	Frequency	Percentage
Extremely important	81	66.9
Very important	32	26.4
Moderately important	8	6.6
Not at all important	0	0
Make Rent and Mortgage Payments (*n* = 98)	Frequency	Percentage
Extremely important	42	56.4
Very important	33	36.1
Moderately important	23	7.4
Not at all important	0	0
Change in Spending Habits (*n* = 85)	Frequency	Percentage
Extremely important	44	51.7
Very important	31	36.4
Moderately important	10	11.7
Not at all important	0	0
Job Loss (*n* = 77)	Frequency	Percentage
Extremely important	22	28.5
Very important	19	24.6
Moderately important	16	20.7
Not at all important	20	25.9

**Table 7 ijerph-21-00374-t007:** Changes in spending habits during the pandemic.

	Frequency	Percentage
Fewer visits to retail stores	75	58.5
Shopping more from variety stores	109	85.1
Looking for discounts	98	76.5
Depending more on online shopping	68	53.1
Stock up on essential foods	83	64.8

**Table 8 ijerph-21-00374-t008:** Social and emotional concerns of Iraqi refugees during the pandemic.

Loneliness and Social Isolation (*n* = 104)	Frequency	Percentage
Extremely important	53	50.9
Very important	30	28.8
Moderately important	21	20.1
Not at all important	0	0
Relationship with Spouse and Children (*n* = 86)	Frequency	Percentage
Extremely important	49	56.9
Very important	21	24.4
Moderately important	16	18.6
Not at all important	0	0
Change in Spending Habits (*n* = 85)	Frequency	Percentage
Extremely important	44	51.7
Very important	31	36.4
Moderately important	10	11.7
Not at all important	0	0

## Data Availability

In adherence to research ethics approval, we are constrained from openly disclosing the raw data.

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
