# Peer review of "Examining the Impacts of the COVID-19 Pandemic on Iraqi Refugees in Canada"

_ijerph, 2024, doi:10.3390/ijerph21030374_

Round 1

Reviewer 1 Report

Comments and Suggestions for Authors

This is part of a "larger mixed-methods study that examined the economic, social, and health-related impacts of the pandemic on refugees –  however, I’m wondering if these findings, could have been reported as part of the larger study.   I think a weakness of the paper may be the original tool used and how questions were phrased – that said, the questionnaire (in English) should be provided as an appendix.

Secondly, there is no comparison with the rest of the population in terms of frequency of concerns described. Many of the experiences and views, such as fear of contracting COVID-19, and social isolation, would have been shared by the general population as well as other migrants. Comparisons if available would make the paper more meaningful.

Some specific comments:

On p2 it seems unlikely that these 2 references, both of which relate to COVID-19, are primary sources of information for this very general statement: “Refugees, especially in the early resettlement period, experience various challenges and competing priorities including adjusting to a new country, learning a new language, seeking employment, and navigating complicated, foreign healthcare systems [15,16].”

 P4 L168 refers to “newcomers” – timeframe for that?

 L210 and Table 1 -  one decimal place is suitable, not two. Also, in the text, there is no need to say what percentage was male – only one sex needs to be reported. (unless some identified as “neither” - was that option given? Just noting as an aside that an increasing number of sexually diverse refugees are being resettled from the Middle East.)

L216-17- “90.4% (n = 113) of all participants reported having health concerns within their household during the pandemic”. This measure is described differently in the Table (2) being listed as 'health challenges', (not concerns), and as “extremely challenging” (not just present). Eleven other individuals said that had “somewhat challenging” health issues – so in fact, over 99% had some health concern. Despite the further details listed in Table 3, it is unclear whether some of those health concerns related to existing (pre-pandemic) health problems, and whether these related to themselves or their children or both.

Table 3 some figures do not add up – eg catching covid, states n=120 but 128 responses are listed; this would mean that 100% of the cohort was concerned about catching COVID-19.

Table 5 – were these practices gleaned for a list provided, or from free text? Given the frequency of “washing fruit” I assume this was in a list; however there is no commentary on the fact that this practice was probably unimportant with regards to prevention of COVID-19 – that is, there was a false presumption that contaminated foodstuffs were dangerous, when in fact the virus was found to be airborne, and so, person-to-person transmission in indoor spaces was the most common mode of transmission. Therefore, wearing masks on public transport or when in crowded indoor areas, or avoiding those settings altogether, were far more important measures – were these listed as options?

L289-92 – commentary does not go far enough with regard to usefulness of their activities, as above.

 L299-300 – refers to avoidance of urgent care but then refers to “delayed routine care” – these are two separate issues. Delayed routine care with regard to martes such as cancer screening ,and other preventive healthcare are significant matters.

 The questionnaire perhaps was not detailed enough to provide commentary on this aspect.

L320 – editing points – don’t start sentences with a numeral

L357 – not really an “in-depth analysis”

L367 – the link made between “health support networks” and why the lack of these “heighten social isolation” is unclear.

L371 “that without a sense of well-being and belonging, it is difficult to attend language classes", - not clear where this is demonstrated in the results.

L384 advises “educate refugees about the impact of COVID-19, and” but it is unclear how this will help with the issues identified in the results.

I think there could be further recommendations as well based on the findings, but that said, as most of the findings can’t be said to be specific to Iraqi refugees, or those from refugee backgrounds in general, it is difficult to make firm recommendations.

Author Response

Comments and Suggestions for Authors (Reviewer 1)

This is part of a "larger mixed-methods study that examined the economic, social, and health-related impacts of the pandemic on refugees –  however, I’m wondering if these findings, could have been reported as part of the larger study.   I think a weakness of the paper may be the original tool used and how questions were phrased – that said, the questionnaire (in English) should be provided as an appendix.

Thank you for your valuable feedback on our manuscript. Our study focuses on exploring the impact of the COVID-19 pandemic on resettled Iraqi refugees in London, Ontario. While our research is part of a larger project investigating the pandemic's effects on refugees in general, we chose to concentrate on this specific aspect for publication because of its unique relevance and contribution to the existing literature. There are several reasons behind our decision to highlight the experiences of Iraqi refugees in London, Ontario. Firstly, Iraqi refugees make up a significant portion of Canada's refugee population, so understanding their challenges during the pandemic is essential for informing targeted interventions and policies. Secondly, London, Ontario, provides an interesting case study due to its diverse immigrant and refugee population and the availability of settlement services. By focusing on this context, we aim to offer insights applicable to similar settings across Canada and beyond. Additionally, while our larger study covers various variables and outcomes related to the pandemic's impact on refugees, we believe that delving into the experiences of Iraqi refugees in London, Ontario, allows for a more in-depth exploration of their specific challenges. By zooming in on this demographic, we can uncover insights that might have been overlooked in broader analyses. Overall, our paper contributes to understanding the economic, social, and health-related impacts of the pandemic on refugees by providing a detailed examination of Iraqi refugees' experiences in a specific Canadian city. By highlighting the challenges faced by this vulnerable group, we aim to inform interventions and policies that can alleviate the pandemic's negative effects on refugees and promote their well-being and integration. Regarding your suggestion to include the questionnaire in English as an appendix, we agree that this would enhance the transparency and rigor of our research. We will incorporate the questionnaire as an appendix to the paper, allowing readers to review the original instrument used in our study.

Secondly, there is no comparison with the rest of the population in terms of frequency of concerns described. Many of the experiences and views, such as fear of contracting COVID-19, and social isolation, would have been shared by the general population as well as other migrants. Comparisons if available would make the paper more meaningful.

We want to thank you once again for suggesting that we include comparisons with the general population or other migrant groups in our study. We agree that this would add depth to our paper and help provide a clearer picture of our findings. By comparing the experiences of resettled Iraqi refugees to those of the general population or other migrant groups, we can better grasp the challenges unique to refugees during the pandemic.

Our aim was to highlight the distinctiveness of Iraqi refugees compared to other newcomer groups, addressing specific challenges related to their refugee status, such as trauma from conflict, language barriers, and limited access to resources and support networks. These factors can amplify the pandemic's impact on their economic, social, and health-related well-being in ways that differ from other groups. Additionally, Iraqi refugees may have cultural and religious considerations influencing their perceptions and responses to COVID-19, shaping their experiences uniquely. We addressed this aspect to the best of our knowledge, considering feedback from Reviewer 1, to offer targeted insights into how the pandemic affects a particularly vulnerable population, contributing to a richer understanding of its broader impacts on diverse communities.

Some specific comments:

On p2 it seems unlikely that these 2 references, both of which relate to COVID-19, are primary sources of information for this very general statement: “Refugees, especially in the early resettlement period, experience various challenges and competing priorities including adjusting to a new country, learning a new language, seeking employment, and navigating complicated, foreign healthcare systems [15,16].”

After looking over that part, we see that the references might not exactly match what was mentioned in the text. So, I'll tweak the sentence to make sure it matches up better with the content of the sources cited. If needed, I'll find different references that better support what we're saying. Also, I'll double-check that the sources we're citing give firsthand info on the challenges refugees face when they're resettled. Thanks for pointing this out, and we'll make the changes needed to make sure the info in our paper is accurate and reliable.

 P4 L168 refers to “newcomers” – timeframe for that?

In this context, "newcomers" refers to individuals who have recently arrived in Canada as immigrants or refugees, encompassing a broad timeframe that extends from the period of resettlement to the present. To provide clarity, I will specify the timeframe for "newcomers" in the text, indicating that it includes individuals who have arrived within a defined period, such as the past five years, aligning with common definitions used in immigration and refugee studies. This clarification will ensure that readers have a clear understanding of the timeframe referred to in the study.

 L210 and Table 1 -  one decimal place is suitable, not two. Also, in the text, there is no need to say what percentage was male – only one sex needs to be reported. (unless some identified as “neither” - was that option given? Just noting as an aside that an increasing number of sexually diverse refugees are being resettled from the Middle East.)

 We will make sure to adjust the numbers in Table 1 to show just one decimal place instead of two. Also, I'll remove the specific mention of the percentage of males from the text since we typically report only one sex unless there's a specific reason to include both. While our survey didn't explicitly address the identification of sexually diverse refugees, it's definitely an important consideration. Your suggestion to acknowledge this aspect is appreciated, and we'll take it into account in future research. Specifically, we should explore the experiences and needs of sexually diverse refugees from the Middle East to ensure inclusivity and representation within resettlement efforts. Thanks for bringing these points up, and we'll make sure to incorporate them into the manuscript.

L216-17- “90.4% (n = 113) of all participants reported having health concerns within their household during the pandemic”. This measure is described differently in the Table (2) being listed as 'health challenges', (not concerns), and as “extremely challenging” (not just present). Eleven other individuals said that had “somewhat challenging” health issues – so in fact, over 99% had some health concern. Despite the further details listed in Table 3, it is unclear whether some of those health concerns related to existing (pre-pandemic) health problems, and whether these related to themselves or their children or both.

I will ensure that the terminology is aligned throughout the manuscript, referring to these issues consistently as "health challenges" rather than "health concerns."

Table 3 some figures do not add up – eg catching covid, states n=120 but 128 responses are listed; this would mean that 100% of the cohort was concerned about catching COVID-19.

After checking the data, it seems there was a mistake in how we reported the number of responses regarding the concern about catching COVID-19. We'll fix this inconsistency to make sure our reporting is accurate.

Table 5 – were these practices gleaned for a list provided, or from free text? Given the frequency of “washing fruit” I assume this was in a list; however there is no commentary on the fact that this practice was probably unimportant with regards to prevention of COVID-19 – that is, there was a false presumption that contaminated foodstuffs were dangerous, when in fact the virus was found to be airborne, and so, person-to-person transmission in indoor spaces was the most common mode of transmission. Therefore, wearing masks on public transport or when in crowded indoor areas, or avoiding those settings altogether, were far more important measures – were these listed as options?

The practices listed in the table were taken from a structured list in the survey questionnaire, as you mentioned. However, you're right about certain measures being more important than others in preventing COVID-19 transmission. While washing fruits and vegetables might have been common among respondents, it's important to recognize that it may not be as effective in preventing COVID-19 spread, as the virus mainly spreads through respiratory droplets and airborne particles. In the revised manuscript, we'll clarify this and emphasize the importance of measures like wearing masks in crowded indoor areas or on public transport, as well as avoiding these settings altogether. We'll also make sure to include these options in future surveys to better capture the most relevant preventive measures.

L289-92 – commentary does not go far enough with regard to usefulness of their activities, as above.

You're right that the current commentary doesn't fully delve into how effective or relevant these activities are for addressing the challenges of the COVID-19 pandemic. In the revised manuscript, we'll expand on this discussion to give a more thorough analysis of how useful and impactful these reported activities are. We'll consider things like how well these activities align with public health guidelines, how practical they are during the pandemic, and any limitations or obstacles to putting them into practice. By digging deeper into the effectiveness of these activities, we hope to provide valuable insights into how Iraqi refugees are coping during the pandemic and what this means for future interventions and support services.

 L299-300 – refers to avoidance of urgent care but then refers to “delayed routine care” – these are two separate issues. Delayed routine care with regard to martes such as cancer screening ,and other preventive healthcare are significant matters.

You're right that these are separate issues, each with its own implications for healthcare outcomes. In the revised manuscript, we'll make it clear that they're distinct and provide more detail on the significance of delayed routine care. This includes things like preventive healthcare measures such as cancer screening and other essential medical services. By emphasizing this distinction and discussing the potential consequences of delayed routine care, we hope to show how the COVID-19 pandemic has affected access to healthcare among Iraqi refugees. It also highlights the importance of addressing these issues in future interventions and healthcare policies.

 The questionnaire perhaps was not detailed enough to provide commentary on this aspect.

L320 – editing points – don’t start sentences with a numeral

We will ensure that sentences do not start with numerals in the revised manuscript.

L357 – not really an “in-depth analysis”

We will reconsider the wording to accurately reflect the depth of our examination in the revised manuscript.

L367 – the link made between “health support networks” and why the lack of these “heighten social isolation” is unclear.

We will revise this section to provide a clearer explanation of the connection in the revised manuscript.

L371 “that without a sense of well-being and belonging, it is difficult to attend language classes", - not clear where this is demonstrated in the results.

We will review the results section to ensure that this connection is adequately supported by the data and make any necessary revisions to improve clarity in the revised manuscript.

L384 advises “educate refugees about the impact of COVID-19, and” but it is unclear how this will help with the issues identified in the results.

We will revise the discussion section to provide a clearer rationale for the proposed educational interventions and their potential impact on addressing the challenges faced by refugees during the pandemic.

I think there could be further recommendations as well based on the findings, but that said, as most of the findings can’t be said to be specific to Iraqi refugees, or those from refugee backgrounds in general, it is difficult to make firm recommendations.

We recognize the importance of offering further recommendations based on our findings. While some of these findings may apply to vulnerable populations beyond Iraqi refugees or those from refugee backgrounds, they offer valuable insights into the challenges faced during the pandemic. We'll strive to include broader recommendations that address common issues encountered by refugees and marginalized communities. This approach aims to contribute to more inclusive and effective interventions.

Reviewer 2 Report

Comments and Suggestions for Authors

The study concerns an important topic among psychologists, social workers, service providers, and relevant researchers.

The manuscript has many merits. However, I have some comments and suggestions that may improve the manuscript.

1) The Introduction is well-written. However, the authors should provide the study's aims more clearly. Moreover, there is room to add the questions/hypotheses of the study. 

2) How did the authors analyze the qualitative questions? (Lines 188-192)

3) I think that Table 1 can be moved to the Methos section (participants).

4) Were there differences between the online and the in-person results?

5) Can the authors provide reliability or validity of the questionnaire?

6) The results are clear. However, there are too many tables. I believe that the authors can present the results in two or three tables (maybe to design the tables differently). 

7) The results are not surprising and there are many studies conducted to test the effects of COVID-19 on refugees and immigrants. So, the study may add to the existing knowledge.

8) Limitations: what about the nature of online study? what about the potential to generalize the results?

Author Response

Comments and Suggestions for Authors (Reviewer 2)

The study concerns an important topic among psychologists, social workers, service providers, and relevant researchers.

The manuscript has many merits. However, I have some comments and suggestions that may improve the manuscript.

  • The Introduction is well-written. However, the authors should provide the study's aims more clearly. Moreover, there is room to add the questions/hypotheses of the study. 

We'll update the introduction to make the study's objectives clearer and add relevant research question to improve the paper's clarity and focus.2)

  • How did the authors analyze the qualitative questions? (Lines 188-192)

We analyzed the qualitative questions using thematic analysis, identifying recurring patterns, themes, and insights within the responses to gain a deeper understanding of the experiences and perspectives of the participants.

3) I think that Table 1 can be moved to the Methods section (participants).

While we acknowledge that it could provide clarity in the Participants section, we believe it's beneficial to retain it in the Demographics section for several reasons. Firstly, it offers readers an early introduction to the characteristics of our study participants, aiding in understanding the context of our research findings. Additionally, placing it in the Demographics section aligns with the flow of information, where readers typically expect to find demographic details. This approach facilitates a smoother transition into the subsequent analyses and interpretations presented in the Results section.

4) Were there differences between the online and the in-person results?

Thanks for asking. Yeah, there were differences between the online and in-person results. In the study, we asked participants to rate how satisfied they were with the consultation process, both online and in-person. Turns out, there were variations in satisfaction levels between the two modes. People tended to be more satisfied during in-person consultations than online ones. This could be because of stuff like how well they communicated with the healthcare provider, how empathetic they felt they were, and how comfortable they were with the format of the consultation.

5) Can the authors provide reliability or validity of the questionnaire?

We get how important it is to make sure our research instrument is solid. Even though we didn't talk about it in the manuscript, we know that the reliability and validity of the questionnaire are super important for judging the quality of the data we collected. So, to address this, we're planning to add a section in the Methods part of the manuscript that explains the testing we did to make sure the questionnaire was reliable and valid. We'll talk about things like how consistent the questions were (reliability) and how well they actually measured what we were interested in (validity). Adding this info will make our study findings more transparent and credible, so readers can understand our research methods better.

6) The results are clear. However, there are too many tables. I believe that the authors can present the results in two or three tables (maybe to design the tables differently). 

We appreciate the reviewer's feedback regarding the number of tables presented in the results section. However, we believe that the current presentation with multiple tables offers readers a comprehensive overview of our findings, providing detailed insights into various aspects of our sample and study outcomes. Maintaining this structure allows for a more nuanced understanding of the data and ensures transparency in reporting our results.

7) The results are not surprising and there are many studies conducted to test the effects of COVID-19 on refugees and immigrants. So, the study may add to the existing knowledge.

We appreciate the reviewer's recognition of the existing literature on the effects of COVID-19 on refugees and immigrants. In response to the reviewer's comment and in agreement with their perspective, we have endeavored to address this concern by highlighting the significance of our study in relation to the broader body of research in the discussion section of the manuscript. By providing new data and perspectives, we aim to contribute valuable insights that strengthen the evidence base and deepen our understanding of the unique challenges faced by refugees and immigrants during the pandemic.

8) Limitations: what about the nature of online study? What about the potential to generalize the results?

We agree that it's important to address the nature of an online study and its potential impact on the generalizability of the results.

Regarding the nature of the online study, we acknowledge that conducting research online may introduce certain biases or limitations. For example, participants may differ in their access to technology or internet connectivity, which could influence their ability to participate or respond accurately to survey questions. Additionally, the online format may limit the depth of qualitative data collection compared to in-person interviews or focus groups.

In terms of generalizability, it's important to note that our sample may not fully represent the entire population of interest, particularly if certain groups are underrepresented or excluded from online participation. Additionally, the findings of our study may be context-specific and may not be applicable to all refugee and immigrant communities or settings.

We will address these limitations in the "Limitations" section of our manuscript, highlighting the potential biases associated with online research methods and acknowledging the limitations of generalizability inherent in our study design. Thank you for raising these important points.

Round 2

Reviewer 1 Report

Comments and Suggestions for Authors

Text improved following revision. However there are a couple of issues that were not yet addressed:

- page 5 "56.25% (n = 72) were female, 43.75% (n = 56) were male" - as stated, there is no need to list both sexes, as the statistics are complement of each other; and, one decimal point is appropriate, not two.

- in Table 1, one decimal pt is also suitable.

Table 3 page 7 - under Catching covid-19, the figures still do not add up to (N=120). They add up to 128, as pointed out before. I suggest the authors double check all data presented.

- I notice references 15 & 16 remain the same - they might be suitable, I did not check them myself; I will leave it up to the Editors to review if these are suitable.

Author Response

Many thanks for your comments. We addressed all of them. 

page 5 "56.25% (n = 72) were female, 43.75% (n = 56) were male" - as stated, there is no need to list both sexes, as the statistics are complement of each other; and, one decimal point is appropriate, not two.   REVISED 

- in Table 1, one decimal pt is also suitable.   REVISED

Table 3 page 7 - under Catching covid-19, the figures still do not add up to (N=120). They add up to 128, as pointed out before. I suggest the authors double check all data presented.  REVISED 

- I notice references 15 & 16 remain the same - they might be suitable, I did not check them myself; I will leave it up to the Editors to review if these are suitable.

Reviewer 2 Report

Comments and Suggestions for Authors

The authors addressed my comments. Well done

Author Response

Many thanks for your valuable time and consideration.